# Protein expression-based classification of gastric cancer by immunohistochemistry of tissue microarray

Chong Zhao[1,2☯], Zhiqiang Feng[1,2☯], Hongzhen He[1,2], Dan Zang[3], Hong Du[4], Hongli Huang[1,2], Yanlei Du[1,2], Jie He[1,2], Yongjian Zhou[1,2]*, Yuqiang Nie[1,2]*

1 Department of Gastroenterology, Guangzhou First People's Hospital, Medical School, South China University of Technology, Guangzhou, China, 2 Department of Gastroenterology, Guangzhou Digestive Disease Center, Guangzhou First People's Hospital, Guangzhou Medical University, Guangzhou, China, 3 Department of Pathology, The First Affiliated Hospital of Xinxiang Medical University, Xinxiang, China, 4 Department of Pathology, Guangzhou First People's Hospital, Medical School, South China University of Technology, Guangzhou, China

☯ These authors contributed equally to this work.
* eynieyuqiang@scut.edu.cn (YN); eyzhouyongjian@scut.edu.cn (YZ)

**Data Availability Statement:** All relevant data are within the paper and its Supporting Information file.

## Abstract

Recently, the Cancer Genome Atlas and Asian Cancer Research Group propose two new classifications system of gastric cancer by using multi-platforms of molecular analyses. However, these highly complicated and cost technologies have not yet been translated into full clinical utility. In addition, the clinicians are expected to gain more guidance of treatment for different molecular subtypes. In this study, we developed a panel of gastric cancer patients in population from Southern China using commercially accessible TMA and immunohistochemical technology. A cohort of 259 GC patients was classified into 4 subtypes on the basis of expression of mismatch repair proteins (PMS2, MLH1, MSH2, and MSH6), E-cadherin and p21 protein. We observed that the subtypes presented distinct prognosis. dMMR-like subtype was associated with the best prognosis, and E-cadherin-a subtype was associated with the worst prognosis. Patients with p21-High and p21-Ligh subtypes had intermediate overall survival. In multivariate analysis, the dMMR-like subtype remained an independent prediction power for overall survival in the model. We described a molecular classification of gastric cancers using clinically applicable assay. The biological relevance of the four subtypes was illustrated by significant differences in prognosis. Our molecular classification provided an effective and inexpensive screening tool for improving prognostic models. Nevertheless, our study should be considered preliminary and carries a limited predictive value as a single-center retrospective study.

## Introduction

Gastric cancer (GC) is the third-leading cause of cancer-related mortality worldwide [1, 2], and more than half of those cases occur in eastern Asian countries [3, 4]. Although curative

**Funding:** This work was supported by the National Natural Science Foundation of China (No. 81871905, No. 81700486), the Natural Science Foundation of Guangdong, China (2020A1515010953), the Science and Technology Program of Guangzhou, Chian (201906010052, 202002030288), and the Fundamental Research Funds for the Central Universities, SCUT (2017BQ113). The funders had no role in study design, data collection and analysis, decision to publish, or preparation of the manuscript.

**Competing interests:** The authors have declared that no competing interests exist.

resection with subsequent adjuvant chemotherapy has been an effective treatment [5–7], the outcome is disappointing with a 5-year survival rate of <40% for patients with II-III stage GC [5, 8]. One of the key reasons for these poor results is that traditional classifications of GC with anatomic sites and histopathology have little therapeutic relevance or predict prognosis [9, 10]. Thus, there is an urgent need to provide effort into the identification of new molecular classifications for developing more specific treatments for GC.

Recent developments in high-throughput sequencing technologies have led to the discovery of new molecular subtypes of GC. As part of The Cancer Genome Atlas (TCGA) project, researchers of the TCGA network recently characterized the genome and proteome of GC based on highly complicated bioinformatics analysis of whole DNA sequencing data, RNA sequencing data, and protein array data [11]. The TCGA study identified 4 genomic subtypes: EBV+ tumors, microsatellite instable (MSI) tumors, genomically stable (GS) tumors, and tumors with chromosomal instability (CIN). Another large-scale study by the Asian Cancer Research Group (ACRG) established four molecular subtypes including MSS/EMT subtype, MSI subtype, MSS/TP53-active subtype, and MSS/TP53-inactive subtype [12]. These classifications can provide molecular subtyping framework for preclinical, clinical and translational studies of GC to find effectively targeted agents in the future. Regarding to the clinical operability, the ACRG proposed some simple methodologies such as immunohistochemistry as alternative. The immunohistochemistry of mismatch-repair proteins will help to define MSI subtype. E-cadherin immunohistochemistry will help to define MSS/EMT subtype. CDKN1A (p21) immunohistochemistry will help to define subtypes of MSS/TP53-active and MSS/TP53-inactive. Currently, several follow-up work of the ACRG study conducted gene expression profiling using different platforms and potential prognostic gene signatures have been developed for the prognosis of GC [13, 14]. Although significant progress has been made in defining various molecular subtypes, effective translation of these complex classifiers into clinical practice depend on their execution for laboratory diagnostic testing and the therapeutic implications.

Since previous study cohorts are mostly from non-China population, the application of these findings on the clinical practice in Chinese cohort is also unclear. In this study we developed a panel of GC patients in population from Southern China using commercially accessible routine diagnostic practice.

## Material and methods

### Patients and tissue samples

The patients were recruited on the basis of the following criteria: histologically confirmed adenocarcinoma of the stomach; surgical resection of primary GC; age ≥ 18 yeas; and complete pathological, surgical, treatment, and follow-up data. We selected 259 patients with histologically confirmed GC who underwent surgery at Guangzhou First People's Hospital (GFPH) between January 2007 and December 2015, and the medical data were designated as the GFPH cohort. Of the 259 patients in GFPH cohort, 136 had received standard adjuvant postoperative chemotherapy (oxaliplatin-based regimen included FOLFOX and XELOX) within one month after surgery, and the others had not due to the financial reasons. None of them had perioperative chemoradiation or neoadjuvant chemotherapy. Tumors were histologically staged according to the 7th edition of the TNM classification by the American Joint Committee on Cancer (AJCC).

### Tissue microarray construction

The tissue microarray (TMA) was constructed from the resection specimens of primary gastric tumors of 259 patients. For the tissue microarray (TMA) analysis, all hematoxylin and eosin-

stained tumor sections were review by one pathologist (Z. D) to define diagnostic areas. Two 2mm-sized cores were obtained from the representative areas of the samples then reembedded in microarray blocks (Beecher Instruments, WI). Each tissue was sampled twice; one core was obtained from the center and the other from the periphery of the tumor. Digital images of TMA immunohistochemically stained slides were obtained via an Aperio Scanscope XT system (Leica Biosystems, Wetzlar, Germany).

## Immunohistochemical staining

Immunohistochemical staining was performed according to the procedure described previously [15, 16]. TMA slides were deparaffinized, rehydrated, and boiled in a pressure cooker filled with a sodium citrate buffer (pH 6.0) for antigen retrieval. After antigen retrieval, the slides were blocked with inhibitor (3% H2O2) for 30 min at 37˚C. Immunohistochemical was performed using the following antibodies and dilutions: MLH1 (1:100, Abcom, Cambridge, USA); PMS2 (1:100, Abcom, Cambridge, USA); MSH6 (3E1, 1:500, Cell Signaling Technology, Beverly, USA); MSH2 (1:200, Abcom, Cambridge, USA); E-cadherin (24E10, 1:400, Cell Signaling Technology, Beverly, USA); p21 Waf1/Cip1, (12D1, 1:50, Cell Signaling Technology, Beverly, USA). All primary antibodies were applied at room temperature for 30 min, followed with a universal biotinylated secondary antibody, 0.05% diaminobenzidine substrate, and haematoxylin counterstain. Tissue samples known to express each marker were used as positive controls. A negative control for every antibody was incubated with preimmune rabbit serum.

## Evaluation of immunostaining

Scoring of the TMA immunohistochemical staining was completed by two independent gastrointestinal pathologists (Z. D. and D. H.) with blind to clinical outcome. In case of discrepant scores between the two observers for each patient, the averaged score was taken into account. Immunohistochemical analysis was conducted according to the system using the parameters described below. An aberrant expression of mismatch repair protein (PMS2/ PMS2/MSH6/ MSH2) was designated as showing complete loss of nuclear staining, whereas tumor cells that showed the presence of nuclear expression, regardless of the proportion or intensity, was classified as normal expression [17]. An abnormal expression of E-cadherin was defined as complete loss of membranous expression or apparently reduced membranous staining (>30%), irrespective of the nuclear or cytoplasmic staining [18]. The expression of the p21 protein was evaluated using a semiquantitative scoring method. Staining was recorded on a scale of 0, 1+ to 3+. Specifically, 0 = negative staining (no nuclear staining of any tumor cells), 1+ = weak expression (nuclear staining of 10% of tumor cells), 2+ = moderate expression (nuclear staining of 10%-25% of tumor cells) moderate, and 3+ = strong expression (nuclear staining of >50% of tumor cells). Take together, 0 and 1+ defined as low expression, 2+ and 3+ defined as high expression.

## Statistical analysis

The correlation of each subtype with overall survival in the GFPH cohort was estimated using Kaplan–Meier plots and log-rank tests. Clinicopathologic features were analyzed for differences within subtypes using the Student t-test, the Chi-square test, or the Fisher exact test. P values less than 0.05 were considered statistically significant. All statistical analysis was completed using SPSS version 18.0 (IBM, Chicago, IL).

### Ethical approval

The current study was approved by the Institutional Review Board at the Guangzhou First People's Hospital, Guangzhou, China (IRB No. K-2018-004-01). To protect personal privacy, identifying information in the electronic database was encrypted. Informed consent was waived by the ethics committee because no intervention was involved and no patient-identifying information was included.

## Results

### Clinical and pathological features of GC patients

The cohort consisted of 156 (60.2%) men and 103 (39.8%) women, with median patient age of 64 (range 26–89) years. The intestinal type, diffuse type, and mixed type by Lauren classification accounted for 33.6%, 49.8%, and 14.7%, respectively. In addition, for tumor histologic differentiation, the poor type was observed with the highest frequency (57.9%) and the high type accounted for 3.9% only. More than half of the tumors were located in the antrum and 22.4% for body, while about 8.8% were located in the cardia and gastroesophageal junction. Noticeably, 137 cases (52.9%) were in stages III according to the 7th edition of the TNM classification by AJCC [19]. 136 patients received adjuvant chemotherapy and all regimens were almost based on oxaliplatin after surgery. The demographic features and clinicopathologic data of the 259 GC patients are summarized in Table 1.

### Protein expression profiles of gastric cancers according to immunohistochemistry results

According to the already proposed molecular classifications by ACRG [12], we investigated several molecular markers using immunohistochemistry (Fig 1), including mismatch repair proteins, E-cadherin and p21 Waf1/Cip1. As showed in Fig 1A, dMMR-like was defined as complete loss of expression in one of PMS2, MLH1, MSH2 or MSH6. Of the 259 patients, 57 showed loss of expression at least one MMR protein on IHC. The most common deficiency identified was MLH1 in 44 patients. PMS2 was identified in 37 patients, MSH6 in 28 patients, and MSH2 in 16 patients. Overall, in the abnormal IHC group, 34 (59.6%) had simultaneous loss of MLH1 and PMS2 expression, 16 (28.1%) had a simultaneous loss of MSH2 and MSH6, and 3 (5.3%) had loss of both MSH6 and PMS2. 10 cases (17.5%) and 8 cases (14.1%) had a loss of only MLH1 and MSH6, respectively. Abnormal expression of E-cadherin, including >30% reduced membranous and absent expression, was found in the 90 out of 259 GC tissues (Fig 1B). Scores of 0 and 1+ were regarded as low expression of p21 and scores of 2+ and 3+ as high expression of p21. Among 259 cases with a semiquantitative scoring of p21 expression, 98 cases were seen high expression of p21 (Fig 1C).

### A four-tier classification of gastric cancers based on protein expression profiles

Based on the previous IHC results and hierarchical clustering analysis of the expression mismatch repair proteins, E-cadherin and p21, we generated a four-tier classification algorithm of GFPH cohort. Out of the cohort (n = 259), 57 cases of dMMR-like phenotype were first found in the total GC samples (Cluster 1); abnormal expression of E-cadherin (Cluster 2) was noted in the MMR-proficient GC samples (n = 75); tissue sample with low p21 expression (Cluster 3) was identified in the normal E-cadherin expression cases (n = 73). Cluster 4 was the rest cases with high p21 expression (n = 54). We named these four groups dMMR-like, E-cadherin-a, p21-Low and p21-High, respectively. Fig 2A illustrates the protein expression-based

**Table 1. Patient characteristics (n = 259).**

| Characteristics | No. |
| --- | --- |
| Age | |
| Age $\geq$ 60 | 155 (59.7) |
| Age < 60 | 104 (40.3) |
| Median age (range) | 64 (26–89) |
| Gender | |
| Male | 156 (60.2) |
| Female | 103 (39.8) |
| Tumor location | |
| Antrum | 175 (67.3) |
| Body | 58 (22.4) |
| Cardia, GEJ | 23 (8.8) |
| whole | 3 (1.3) |
| Tumor size | |
| < 5 cm | 132 (50.9) |
| $\geq$ 5 cm | 127 (49.1) |
| Lauren type | |
| Intestinal | 87 (33.6) |
| Diffuse | 129 (49.8) |
| Mixed | 38 (14.7) |
| Missing | 5 (1.9) |
| pT stage | |
| T1 | 18 (6.9) |
| T2 | 26 (10.0) |
| T3 | 38 (14.7) |
| T4 | 177 (68.3) |
| pN stage | |
| N0 | 98 (37.8) |
| N1 | 37 (14.3) |
| N2 | 56 (21.6) |
| N3 | 68 (26.3) |
| Tumor differentiation | |
| Poor | 150 (57.9) |
| Moderate | 99 (38.2) |
| Well | 10 (3.9) |
| AJCC stage | |
| I b | 37 (14.3) |
| II | 44 (17.0) |
| III | 137 (52.9) |
| IV | 41 (15.8) |
| lymphovascular invasion | |
| Positive | 76 (29.3) |
| Negative | 183 (70.7) |
| Perineural invasion | |
| Positive | 88 (34.0) |
| Negative | 171 (66.0) |
| Adjuvant chemotherapy | |
| Yes | 136 (52.5) |

(*Continued*)

**Table 1.** (Continued)

| Characteristics | No. |
|---|---|
| No | 123 (47.5) |

Abbreviations: EGJ, esophagogastric junction; pT stage, pathological assessment of primary tumor; pN stage, pathological assessment of regional lymph nodes; AJCC, American Joint Committee on Cancer.

classification. Next, we performed a survival analysis and observed obvious differences in overall survival within the 4 subtypes. Of all 259 patients, the median overall survival (OS) was 31 months (95% CI 24.3–35.7 months). As shown in Fig 2B, dMMR-like subtype disclosed the best prognosis. Subsequently, the better prognosis was observed in subtype of p21-High, followed by subtype of p21-Low. The E-cadherin-a subtype revealed the worst prognosis (Log-rank Test, p< 0.0001). Multivariate analysis, adjusted for several covariates, was conducted according to the proportional hazards assumption of the Cox model (Table 2). The dMMR-like subtype remained an independent prediction power for overall survival in the model

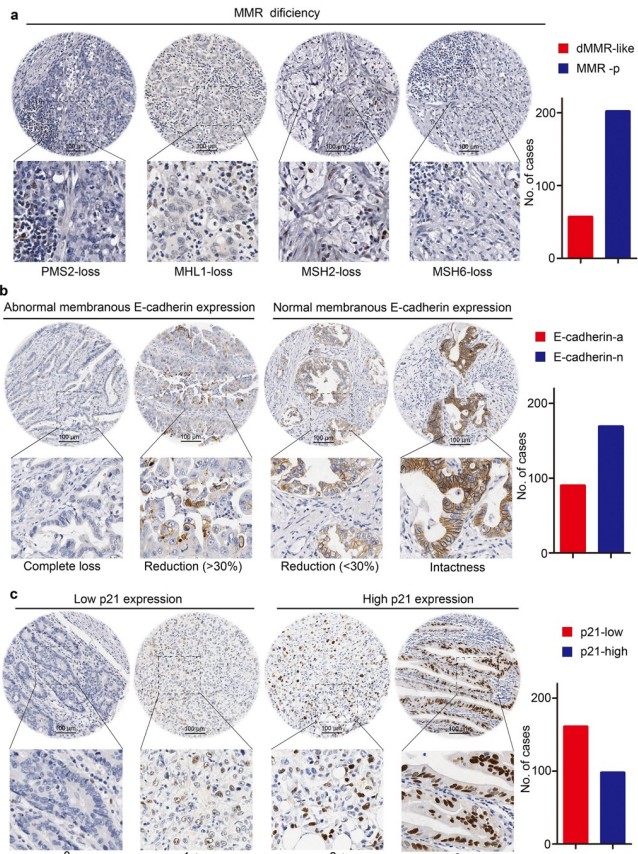

**Fig 1. Representative immunohistochemical images of multiple markers in TMA sections.** (**A**) dMMR-like cancer tissues showed complete loss of PMS2, MLH1, MSH2, and MSH6 staining, but adjacent lymphoid cells or stromal cells were positive for the proteins. (**B**) E-cadherin expression was determined on the basis of membranous staining. (**C**) Typical 0, 1+, 2+, and 3+ scoring of p21 staining. Scale bar = 100μm. The right histograms showed the sample numbers of each marker across cohort, respectively. dMMR-like, mismatch repair protein deficiency like phenotype; MMR-p, mismatch repair protein proficiency; E-cadherin-a, abnormal expression of E-cadherin; E-cadherin-n, normal expression of E-cadherin; p21-low, low expression of p21; p21-high, high expression of p21.

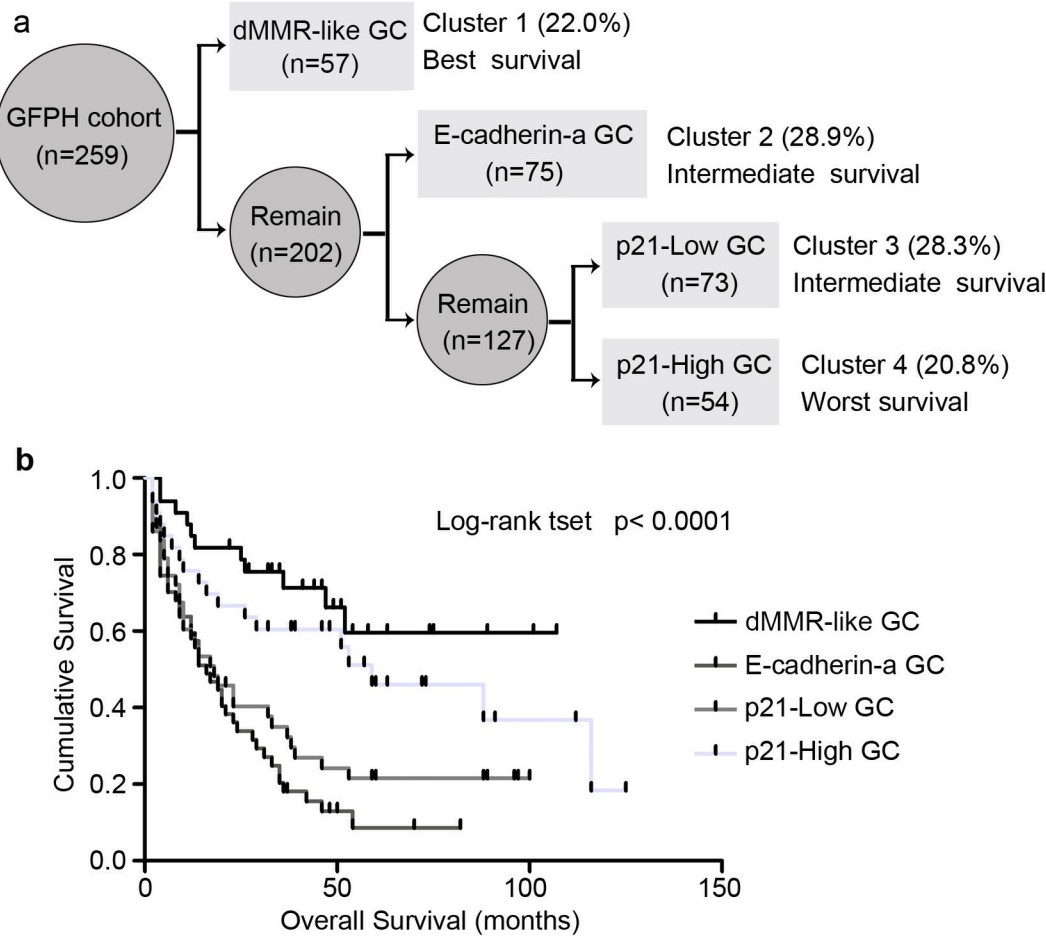

**Fig 2. The four-tier classification of gastric cancer.** (**A**) Schematic diagram of protein expression-based classification. dMMR-like, mismatch repair protein deficiency like phenotype; E-cadherin-a, abnormal expression of E-cadherin; p21-Low, low expression of p21; p21-High, high expression of p21; GC, gastric cancer. (**B**) Survival curves for each of the four subtypes of gastric cancer in patient cohort (Log-rank Test, p< 0.0001).

**Table 2. Multivariable analysis of overall survival.**

| Variable | Hazard Ratio | 95% CI for Hazard Ratio | P |
|---|---|---|---|
| Subtype (dMMR-like vs. Remaining) | 3.556 | 3.556–6.806 | 0.000 |
| Sex (male vs. female) | 1.238 | 0.819–1.871 | 0.312 |
| Location of tumor (antrum vs. nonantrum) | 1.061 | 0.684–1.646 | 0.792 |
| Tumor size (antrum vs. nonantrum) | 1.721 | 1.136–2.605 | 0.010 |
| Lauren type (diffuse vs. intestinal and mixed) | 0.527 | 0.392–1.090 | 0.473 |
| Tumor differentiation (poor vs. remaining) | 0.905 | 0.530–1.546 | 0.715 |
| pT stage (T4 vs. Remaining) | 0.523 | 0.306–0.892 | 0.017 |
| pN stage (N0 vs. Remaining) | 1.071 | 0.661–1.737 | 0.780 |
| Lymphovascular invasion (no vs. yes) | 2.015 | 1.313–3.095 | 0.001 |
| Perineural invasion (no vs. yes) | 1.367 | 0.889–2.103 | 0.154 |

Abbreviations: dMMR-like, mismatch repair protein deficiency like phenotype; pT stage, pathological assessment of primary tumor; pN stage, pathological assessment of regional lymph nodes.

(hazard ratio: 3.556, p<0.001). However, these results should be interpreted with caution because of the low number of events and high number of parameters assessed.

## Clinicopathologic features of the four subtypes of gastric cancers

To investigate whether the molecular classification system has clinical covariates in the GFPH cohort, we summarized the clinicopathological features according to the molecular subtypes. Among these patients, dMMR-like GC constituted 22% of the cohort, 28.9% for E-cadherin-a GC, 28.3% for p21-Low GC and 20.8% for p21-High GC. As shown in Table 3, we found several trends: (i) The dMMR-like subtype tumors were predominantly located in the antrum (82.5%), rarely diagnosed at stage IV (5.3%). (ii) The majority of patients in E-cadherin-a subtype were diagnosed with diffuse-type. (iii) The p21-High subtype tumors had higher proportion of intestinal subtype and lower proportion of poor differentiation than did other subtypes.

## Discussion

Systemic chemotherapy has been showed to improve patient prognosis as a recommended component of resectable GC therapy [5, 20]. However, 30–50% of patients relapse within 5 years after surgery and adjuvant chemotherapy [21, 22]. Preliminary evidence has indicated that variable responses to treatment can be attributed to tumor heterogeneity with regard to molecular alterations [23, 24].

Recently, TCGA and ACRG propose two new classifications system of GC by using multi-platforms of molecular analyses [11, 12]. Although these molecular profiles can potentially be developed into predictive biomarkers of treatment, more work in this area is needed to establish the clinical relevance. Under the guidance and inspiration of ACRG studies [12], we chose several potential biomarkers as follows: Mismatch-repair deficiency like phenotype (dMMR-like) represented a hallmark feature of microsatellite instability (MSI) status [25, 26]; aberrant expression of E-cadherin (E-cadherin-a) was related to epithelial-to-mesenchymal transition (EMT) phenotype [27]; low expression of p21 (p21-Low) was somehow linked to p53 inactivation and high expression of p21 (p21-High) to p53 activation, and then a molecular classification of GC can be established based on hierarchical cluster analysis of the protein profiles. In our cohort, the frequency of dMMR-like subtype gastric adenocarcinoma was 22.0%, close to the MSI subtype in the Cancer Genome Atlas (21.7%) and the Asian Cancer Research Group (22.7%) [11, 12]. In other papers, MSI was found in 5.6–30% of GC [28–31]. Most of studies demonstrated that GCs with MSI were associated with advanced stage, elderly age, less lymph node involvement, intestinal type, and distal location [32–34]. Considering that over 80% of patients were diagnosed at an advanced stage of disease and nearly 70% of GCs were localized in the antrum, high proportion of dMMR-like cases may be acceptable to our study. More importantly, dMMR-like subtype also showed the better prognosis than patients with other subtypes. We next found a higher incidence of E-cadherin-a subtype (28.9%) compared with epithelial–mesenchymal transition (EMT) subtype of the ACRG (15.3%), and this variance may because that the E-cadherin level is not always changed only by EMT. Promoter hyper-methylation has been found to be significantly associated with decreased expression of E-cadherin in gastric cancer, especially in diffuse histological type [35]. However, another study showed a similar frequency of 21%, equally characterized by aberrant E-cadherin expression [18, 36]. Of note they all revealed the worst prognosis as we did. Although a great number of studies were performed on patients with gastric cancer, the prognostic value of E-cadherin for gastric cancer patients remains controversial. Many studies reported that reduced E-cadherin expression was significantly associated with poor overall survival of gastric cancer patients

**Table 3. Clinicopathologic characteristics of the various gastric cancer subtypes.**

| Characteristics | dMMR-like | E-cadherin-a | p21-Low | p21-High | P value |
|---|---|---|---|---|---|
| N (%) | 57 (22.0%) | 75 (28.9%) | 73 (28.3%) | 54 (20.8%) | |
| Age | | | | | 0.6758[a] |
| Median age | 64 | 63 | 66 | 62 | |
| Range | 28–83 | 26–85 | 43–89 | 33–88 | |
| Gender | | | | | 0.6001 |
| Male | 31 (54.4) | 49 (65.3) | 45 (61.6) | 31 (57.4) | |
| Female | 26 (45.6) | 26 (34.7) | 28 (38.4) | 23 (42.6) | |
| Tumor location | | | | | 0.0674 |
| Antrum | 47 (82.5) | 44 (58.7) | 50 (68.5) | 34 (63.0) | |
| Body | 7 (12.3) | 21 (28.0) | 15 (20.5) | 15 (27.8) | |
| Cardia, GEJ | 3 (5.2) | 7 (9.3) | 8 (11.0) | 5 (9.2) | |
| whole | 0 ((0.0) | 3 (4.0) | 0 (0.0) | 0 (0.0) | |
| Lauren type | | | | | 0.0057** |
| Intestinal | 18 (31.6) | 18 (24.0) | 28 (38.3) | 23 (42.6) | |
| Diffuse | 32 (56.1) | 49 (65.3) | 32 (43.8) | 16 (29.6) | |
| Mixed | 5 (8.8) | 8 (10.7) | 12 (16.4) | 13 (24.1) | |
| Missing | 2 (3.5) | 0 (0.0) | 1 (1.4) | 2 (3.7) | |
| pT stage | | | | | 0.0134* |
| T1 | 7 (12.3) | 3 (4.0) | 3 (4.1) | 5 (9.3) | |
| T2 | 7 (12.3) | 3 (4.0) | 5 (6.8) | 11 (20.4) | |
| T3 | 10 (17.5) | 7 (9.3) | 13 (17.8) | 8 (14.8) | |
| T4 | 33 (57.9) | 62 (82.7) | 52 (71.3) | 30 (55.5) | |
| pN stage | | | | | 0.0706 |
| N0 | 29 (50.9) | 18 (24.0) | 26 (35.6) | 25 (46.3) | |
| N1 | 10 (17.5) | 11 (14.7) | 11 (15.1) | 5 (9.2) | |
| N2 | 10 (17.5) | 20 (26.7) | 15 (20.5) | 11 (20.4) | |
| N3 | 8 (14.1) | 26 (34.6) | 21 (28.8) | 13 (24.1) | |
| AJCC stage | | | | | 0.0120 * |
| I b | 13 (22.8) | 7 (9.3) | 6 (8.2) | 11 (20.4) | |
| II | 10 (17.5) | 8 (10.7) | 13 (17.8) | 13 (24.1) | |
| III | 31 (54.4) | 42 (56.0) | 39 (53.4) | 25 (46.3) | |
| IV | 3 (5.3) | 18 (24.0) | 15 (20.6) | 5 (9.2) | |
| Tumor differentiation | | | | | 0.0290* |
| Poor | 37 (64.9) | 52 (69.3) | 37 (50.7) | 24 (44.4) | |
| Moderate | 16 (28.1) | 21 (28.0) | 34 (46.6) | 28 (51.9) | |
| Well | 4 (7.0) | 2 (2.7) | 2 (2.7) | 2 (3.7) | |
| lymphovascular invasion | | | | | 0.1011 |
| Positive | 15 (26.3) | 23 (30.4) | 28 (38.4) | 10 (18.5) | |
| Negative | 42 (73.7) | 52 (69.6) | 45 (61.6) | 44 (81.5) | |
| Perineural invasion | | | | | 0.0130* |
| Positive | 16 (28.1) | 20 (26.7) | 36 (48.3) | 16 (29.6) | |
| Negative | 41 (71.9) | 55 (73.3) | 37 (51.7) | 38 (70.3) | |

Abbreviations: dMMR-like, mismatch repair protein deficiency like phenotype; E-cadherin-a, abnormal expression of E-cadherin; p21-low, low expression of p21; p21-high, high expression of p21.

[a]One-way analysis of variance (ANOVA) test was used. Chi-square test was used for all other variables.

[37–39], while quite a few researchs concluded that E-cadherin was not a prognostic factor for survival [40–42]. Interestingly, E-cadherin low-expression usually had an unfavorable impact on OS in Asian patients.

After exclusion of the dMMR-like and E-cadherin-a clusters, we identified two distinct subgroup of cases, characterized by either low or high p21 expression. Since induction of p21 expression levels provides direct evidence for activation of the p53 protein, high expression of p21 is closely related to p53 activation and low expression of p21 to p53 inactivation [43, 44]. Regarding to the frequency and prognosis, those two subsets were in accordance with the MSS/TP53+ and MSS/TP53- type described by the ACRG, respectively [12]. Apparently, we proposed a simplified algorithm that is also able to exhibit the clinicopathologic features and survival trends for GC patients from Southern China, using commercially accessible immunohistochemical techniques.

Many studies have investigated correlations between their molecular subtypes and clinicopathological features, including male predominance, preferential location, and predominance of WHO classification [12, 18, 36, 45, 46]. In the Asian Cancer Research Group, MSI subtype was predominantly associated with early stage, antrum location, and intestinal phenotype. We confirmed that dMMR-like subtype occurred predominantly in the antrum (82.5%, $P = 0.0674$), which is similar to the data reported in ACRG. However, prominent early stage or intestinal type was not seen in these cases. Consistent with ACRG study, the majority of the subjects in E-cadherin-a subtype were diagnosed with diffuse-type ($>65\%$, $P = 0.0057$) at stage III/IV ($>80\%$, $P = 0.0120$). However, these differences were partially statistically significant; a likely consequence was seen in some subgroups because of the limitation on the number of patient cases. Therefore an investigation that ought to be tested in larger studies.

Our multivariate analysis has some limitations. In particular, some established prognostic factors of GC prognosis, notably Lauren type, tumor differentiation, pN stage, and perineural invasion, did not reach significance may have been because these factors are co-linear with our classification in this study. Thus, the significance and robustness of the signature as a prognostic classification requires further confirmation with large prospective patient cohorts. In addition, many of the patients in this cohort have not received adjuvant chemotherapy, so our classifications could not predict the chemotherapeutic responses precisely.

In summary, we demonstrate that clinical significance of the four subtypes of GC described here is supported by its significant correlation with clinical outcomes. Given the broad similarities with previous molecular classifications, our GC stratification based on immunohistochemical analysis is reproducible, robust and may represent an effective and inexpensive screening tool for individualized treatment of GC.

## Supporting information

**S1 Dataset.**
(XLSX)

## Author Contributions

**Conceptualization:** Chong Zhao, Zhiqiang Feng, Yuqiang Nie.

**Data curation:** Chong Zhao, Zhiqiang Feng, Hongzhen He.

**Formal analysis:** Chong Zhao, Zhiqiang Feng, Hong Du, Hongli Huang.

**Funding acquisition:** Chong Zhao, Hongli Huang, Yuqiang Nie.

**Investigation:** Dan Zang, Yanlei Du.

**Methodology:** Jie He.

**Project administration:** Yongjian Zhou.

**Resources:** Hongzhen He, Hong Du.

**Software:** Dan Zang.

**Supervision:** Jie He, Yongjian Zhou.

**Validation:** Yanlei Du.

**Visualization:** Hongli Huang.

**Writing – original draft:** Chong Zhao, Zhiqiang Feng.

**Writing – review & editing:** Yuqiang Nie.

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
