## [Decision Letter · Decision Letter 0]

21 Oct 2019

PONE-D-19-24376

Protein expression profiling identifies molecular subtypes of gastric cancer and predicts adjuvant chemotherapeutic benefit

PLOS ONE

Dear Dr. Nie,

Thank you for submitting your manuscript to PLOS ONE. After careful consideration, we feel that it has merit but does not fully meet PLOS ONE’s publication criteria as it currently stands. Therefore, we invite you to submit a revised version of the manuscript that addresses the points raised during the review process.

(1) Title. Because the Title of this manuscript is not supported by results, you are advised to modify the Title.

(2) Definition of mismatch repair deficiency (MMR-d). The authors defined MMR-d as “complete loss of expression in one of PMS2, MLH1, MSH2 or MSH6”; however, this definition is not a general definition in the global scientific community. You need to weaken their claim by using “MMR-d-like phenotype” or similar description.

(3) Multi-variate analysis. You are advised to discuss the results of multivariate analysis as weak point in the abstract and discussion.

(4) Other issues. You need to address other issues pointed out by Reviewer 1.

We would appreciate receiving your revised manuscript by Dec 05 2019 11:59PM. To enhance the reproducibility of your results, we recommend that if applicable you deposit your laboratory protocols in protocols.io, where a protocol can be assigned its own identifier (DOI) such that it can be cited independently in the future. For instructions see: http://journals.plos.org/plosone/s/submission-guidelines#loc-laboratory-protocols

We look forward to receiving your revised manuscript.

Kind regards,

Masaru Katoh, M.D., Ph.D.

Academic Editor

PLOS ONE

Journal Requirements:

2) Please ensure that the IRB approval number stated in the following sentence is the correct and corresponds to the number displayed on the ethics documentation provided in previous correspondence:

'The current study was approved by the Institutional Review Board at the Guangzhou First People’s Hospital, Guangzhou, China (IRB No. K-2018-004-01).'

Thank you for your attention to this request.

3) We note that you have included the phrase “data not shown” in your manuscript. Unfortunately, this does not meet our data sharing requirements. PLOS does not permit references to inaccessible data. We require that authors provide all relevant data within the paper, Supporting Information files, or in an acceptable, public repository. Please add a citation to support this phrase or upload the data that corresponds with these findings to a stable repository (such as Figshare or Dryad) and provide and URLs, DOIs, or accession numbers that may be used to access these data. Or, if the data are not a core part of the research being presented in your study, we ask that you remove the phrase that refers to these data.

4) Please include captions for your Supporting Information files at the end of your manuscript, and update any in-text citations to match accordingly. Please see our Supporting Information guidelines for more information: http://journals.plos.org/plosone/s/supporting-information.

Reviewers' comments:

Reviewer's Responses to Questions

**Comments to the Author**

1. Is the manuscript technically sound, and do the data support the conclusions?

Reviewer #1: Partly

2. Has the statistical analysis been performed appropriately and rigorously? 

Reviewer #1: No

3. Have the authors made all data underlying the findings in their manuscript fully available?

Reviewer #1: No

4. Is the manuscript presented in an intelligible fashion and written in standard English?

Reviewer #1: Yes

5. Review Comments to the Author

Reviewer #1: The authors developed a panel of immunohistochemical technology in a cohort of 259 GC patients and classified GC into 4 subtypes on the basis of expression of mismatch repair proteins (PMS2, MLH1, MSH2, and MSH6), E-cadherin and p21 protein. They observed that the subtypes presented distinct prognosis and corresponded to the molecular classifications previously reported. MMR-d subtype was associated with the best prognosis, and E-cadherin-a subtype was associated with the worst prognosis. Patients with p21-High and p21-Ligh subtypes had intermediate overall survival. Patients with the p21-High subtype most benefitted from adjuvant chemotherapy and those with the MMR-d subtype had the least benefit from adjuvant chemotherapy. There are several issues to be considered.

Major

1. The title of the manuscript is “Protein expression profiling identifies molecular subtypes of gastric cancer and predicts adjuvant chemotherapeutic benefit”. However, there are many cases who have not received adjuvant chemotherapy (N=123, 47.5%) and judging benefits of chemotherapeutic effects does not seem simple. So, it is better to delete those sentence from the title of the manuscript and more detailed statistical analyses are required to propose the authors’ hypothesis.

2. The main limitations of the present study are low and no significance observed in multivariate survival analyses in Table 2.

2. In Introduction, the authors described too much on gastric cancer incidences. Instead of this, it would better to add why the authors selected the antibodies used for this study.

Minor

1. In page 5, TMA construction, “Two 2-mm-thick cores” can be replaced by “Two 2mm-sized cores”

2. The numbers of dMMR cases are more frequent than previously reported. Please explain the reason in Discussion.

3. The authors described that “Digital images of TMA immunohistochemically stained slides were obtained via an Aperio Scanscope XT system (Leica Biosystems, Wetzlar, Germany)”. Then, why the authors did not use Aperio software to interpret them? It would be better to get the IHC data by digital image analyses instead of pathologists to reduce inter-observer variability.

4. In abstract, “corresponded to the molecular classifications previously reported” sentence seems inappropriate based on authors’ results.

6. PLOS authors have the option to publish the peer review history of their article (what does this mean?). If published, this will include your full peer review and any attached files.

Reviewer #1: No

---

## [Author Response · Author response to Decision Letter 0]

22 Apr 2020

Editor：

(1) Title. Because the Title of this manuscript is not supported by results, you are advised to modify the Title.

Response: Thanks for pointing this out. We have now modified the title of our manuscript to more reflecting the findings as follow: “A protein expression-based classification of gastric cancer by the Tissue Array Method immunohistochemistry”

 (2) Definition of mismatch repair deficiency (MMR-d). The authors defined MMR-d as “complete loss of expression in one of PMS2, MLH1, MSH2 or MSH6”; however, this definition is not a general definition in the global scientific community. You need to weaken their claim by using “MMR-d-like phenotype” or similar description.

Response: Thanks for your thoughtful suggestion. We have now replaced “MMR-d” with “dMMR-like” or “dMMR-like phenotype” 

 (3) Multi-variate analysis. You are advised to discuss the results of multivariate analysis as weak point in the abstract and discussion.

Response: Following your suggestion, we have now discussed the multivariate analysis results in the abstract and discussion sections (Page 2, lines 14-21; Page 17, lines 19-22; Page 18, lines 1-2).

(4) Other issues. You need to address other issues pointed out by Reviewer 1.

Response: Following the direction from the reviewer 1, we have now substantially revised this manuscript by addressing the main issues raised.

Reviewer #1: 

Major

1. The title of the manuscript is “Protein expression profiling identifies molecular subtypes of gastric cancer and predicts adjuvant chemotherapeutic benefit”. However, there are many cases who have not received adjuvant chemotherapy (N=123, 47.5%) and judging benefits of chemotherapeutic effects does not seem simple. So, it is better to delete those sentence from the title of the manuscript and more detailed statistical analyses are required to propose the authors’ hypothesis.

Response: Thanks for pointing this out. Following your suggestion, we have now eliminated the part about the chemotherapeutic benefit in the manuscript.

2. The main limitations of the present study are low and no significance observed in multivariate survival analyses in Table 2.

Response: Thanks for pointing this out. We have discussed the results of multivariate analysis as weak point in the Discussion as follow: “Our multivariate analysis has some limitations. In particular, some established prognostic factors of GC prognosis, notably Lauren type, Tumor differentiation, pN stage, and perineural invasion, did not reach significance may have been because these factors are co-linear with our classification in this study. Thus, the significance and robustness of the signature as a prognostic classification requires further confirmation with large prospective patient cohorts.” (Page 17, lines 19-22; Page 18, lines 1-2)

3. In Introduction, the authors described too much on gastric cancer incidences. Instead of this, it would better to add why the authors selected the antibodies used for this study.

Response: Following your suggestion, the entire Introduction section is carefully edited to reduce the description of gastric cancer incidences and add the reason for selecting the antibodies used. (Page 4, lines 2-6; Page 5, lines 1-6)

Minor

1. In page 5, TMA construction, “Two 2-mm-thick cores” can be replaced by “Two 2mm-sized cores”

Response: Corrected. Thanks.

2. The numbers of dMMR cases are more frequent than previously reported. Please explain the reason in Discussion.

Response: Following your suggestion, we have explained the reason in Discussion as follow: “In other papers, MSI was found in 5.6–30% of GC. Most of studies demonstrated that GCs with MSI were associated with advanced stage, elderly age, less lymph node involvement, intestinal type, and distal location. Considering that over 80 % of patients were diagnosed at an advanced stage of disease and nearly 70% of GCs were localized in the antrum, high proportion of dMMR cases may be acceptable to our study.” (Page 15, lines 20-22; Page 16, lines 1-4)

3. The authors described that “Digital images of TMA immunohistochemically stained slides were obtained via an Aperio Scanscope XT system (Leica Biosystems, Wetzlar, Germany)”. Then, why the authors did not use Aperio software to interpret them? It would be better to get the IHC data by digital image analyses instead of pathologists to reduce inter-observer variability.

Response: Thanks for pointing this out. This is the first time we use Aperio Scanscope XT system. Our technicians and pathologists currently lack training in the application of this Aperio Image Scope software. Although our pathologists have extensive experience in evaluation of IHC staining, the digital image analyses should be used to reduce inter-observer variability in our future studies.

4. In abstract, “corresponded to the molecular classifications previously reported” sentence seems inappropriate based on authors’ results.

Response: Thanks for pointing this out. We have now eliminated this inappropriate sentence.

---

## [Decision Letter · Decision Letter 1]

1 Jun 2020

PONE-D-19-24376R1

A protein expression-based classification of gastric cancer by the Tissue Array Method immunohistochemistry

PLOS ONE

Dear Dr. Nie,

Thank you for submitting your manuscript to PLOS ONE. After careful consideration, we feel that it has merit but does not fully meet PLOS ONE’s publication criteria as it currently stands. Therefore, we invite you to submit a revised version of the manuscript that addresses the points raised during the 2nd review process.

We look forward to receiving your revised manuscript.

Kind regards,

Girijesh Kumar Patel, PhD

Academic Editor

PLOS ONE

Reviewers' comments:

Reviewer's Responses to Questions

**Comments to the Author**

1. If the authors have adequately addressed your comments raised in a previous round of review and you feel that this manuscript is now acceptable for publication, you may indicate that here to bypass the “Comments to the Author” section, enter your conflict of interest statement in the “Confidential to Editor” section, and submit your "Accept" recommendation.

Reviewer #1: All comments have been addressed

Reviewer #2: (No Response)

2. Is the manuscript technically sound, and do the data support the conclusions?

Reviewer #1: Yes

Reviewer #2: Yes

3. Has the statistical analysis been performed appropriately and rigorously? 

Reviewer #1: Yes

Reviewer #2: Yes

4. Have the authors made all data underlying the findings in their manuscript fully available?

Reviewer #1: Yes

Reviewer #2: No

5. Is the manuscript presented in an intelligible fashion and written in standard English?

Reviewer #1: No

Reviewer #2: Yes

6. Review Comments to the Author

Reviewer #1: The manuscript has been revised accordingly and improved. I have several minor comments.

1) In the title, please change "A protein expression-based classification of gastric cancer by the Tissue Array Method immunohistochemistry" to "Protein expression-based classification of gastric cancer by immunohistochemistry of tissue microarray".

2) In Introduction, please add several efforts to replace molecular classifications of gastric cancer using archival tissue instead of fresh tissue such as nanostring (PMID: 29029513) or RNAseq of FFPE tissues (PMID: 31273278).

3) In Discussion, please add limitations of the present study including "Many of the patients in this cohort have not received adjuvant chemotherapy, so our classifications could not predict the chemotherapeutic responses precisely.

Reviewer #2: This is a manuscript that provide molecular classification that is effective and inexpensive screening tool for improving prognostic models.

1. For patient with aberrant expression of mismatch repair protein (PMS2/ PMS2/MSH6/MSH2), did the author further explore if this is somatic or gremline?

2. For patient with aberrant expression of mismatch repair protein (PMS2/ PMS2/MSH6/MSH2), it is helpful to know if they have received immunotherapy or not compared to other groups

Thanks

7. PLOS authors have the option to publish the peer review history of their article (what does this mean?). If published, this will include your full peer review and any attached files.

Reviewer #1: No

Reviewer #2: No

---

## [Author Response · Author response to Decision Letter 1]

17 Aug 2020

Responses to Reviewers

Reviewer #1: The manuscript has been revised accordingly and improved. I have several minor comments.

1) In the title, please change "A protein expression-based classification of gastric cancer by the Tissue Array Method immunohistochemistry" to "Protein expression-based classification of gastric cancer by immunohistochemistry of tissue microarray".

Response：Thanks for pointing this out. We have now modified the title of our manuscript to more reflecting the findings as follow: “Protein expression-based classification of gastric cancer by immunohistochemistry of tissue microarray”

2) In Introduction, please add several efforts to replace molecular classifications of gastric cancer using archival tissue instead of fresh tissue such as nanostring (PMID: 29029513) or RNAseq of FFPE tissues (PMID: 31273278).

Response：Following your suggestion, we have now add several efforts in Introduction to replace molecular classifications of gastric cancer using archival tissue instead of fresh tissue (Page 5, lines 7-12).

3) In Discussion, please add limitations of the present study including "Many of the patients in this cohort have not received adjuvant chemotherapy, so our classifications could not predict the chemotherapeutic responses precisely.

Response：Following your suggestion, we have now add these limitations in Discussion sections (Page 18, lines 9-11).

Reviewer #2: This is a manuscript that provide molecular classification that is effective and inexpensive screening tool for improving prognostic models.

1. For patient with aberrant expression of mismatch repair protein (PMS2/ PMS2/MSH6/MSH2), did the author further explore if this is somatic or gremline?

Response：This is a very good question because experimentation to address this question would potentially provide additional evidence for this study. However, in order to publish these findings and thereby to stimulate more investigations from others in a timely fashion, we would like leave this rather time-consuming experiment for the future.

2. For patient with aberrant expression of mismatch repair protein (PMS2/ PMS2/MSH6/MSH2), it is helpful to know if they have received immunotherapy or not compared to other groups

Response：Thanks for pointing this out. According recent studies, immunotherapy has brought long-lasting tumor remission for gastric cancer, and mismatch repair deficiency (dMMR) as predictive biomarkers that guide the clinical application of immune checkpoint blockade therapies. As to our research, one of the criteria for recruiting patients is that the treatment plan is limited to chemotherapy. None of them had immunotherapy. As more and more gastric cancer patients receive immunotherapy in China, we would like focus on the relationship between dMMR subgroup and novel immunotherapy in our future studies.

---

## [Editor Report · Decision Letter 2]

26 Aug 2020

Protein expression-based classification of gastric cancer by immunohistochemistry of tissue microarray

PONE-D-19-24376R2

Dear Dr. Nie,

We’re pleased to inform you that your manuscript has been judged scientifically suitable for publication and will be formally accepted for publication once it meets all outstanding technical requirements.

Kind regards,

Girijesh Kumar Patel, PhD

Academic Editor

PLOS ONE
---

## [Editor Report · Acceptance letter]

13 Oct 2020

PONE-D-19-24376R2 

Protein expression-based classification of gastric cancer by immunohistochemistry of tissue microarray 

Dear Dr. Nie:

I'm pleased to inform you that your manuscript has been deemed suitable for publication in PLOS ONE. Congratulations! Your manuscript is now with our production department. 

Kind regards, 

on behalf of

Dr. Girijesh Kumar Patel 

Academic Editor

PLOS ONE